# Sparse Polynomial Learning and Graph Sketching

**Murat Kocaoglu**[1*], **Karthikeyan Shanmugam**[1†], **Alexandros G.Dimakis**[1‡], **Adam Klivans**[2*]

[1]Department of Electrical and Computer Engineering, [2]Department of Computer Science
The University of Texas at Austin, USA
*mkocaoglu@utexas.edu, †karthiksh@utexas.edu
‡dimakis@austin.utexas.edu, *klivans@cs.utexas.edu

## Abstract

Let $f : \{-1,1\}^n \to \mathbb{R}$ be a polynomial with at most $s$ non-zero real coefficients. We give an algorithm for exactly reconstructing $f$ given random examples from the uniform distribution on $\{-1,1\}^n$ that runs in time polynomial in $n$ and $2^s$ and succeeds if the function satisfies the *unique sign property*: there is one output value which corresponds to a unique set of values of the participating parities. This sufficient condition is satisfied when every coefficient of $f$ is perturbed by a small random noise, or satisfied with high probability when $s$ parity functions are chosen randomly or when all the coefficients are positive. Learning sparse polynomials over the Boolean domain in time polynomial in $n$ and $2^s$ is considered notoriously hard in the worst-case. Our result shows that the problem is tractable for almost all sparse polynomials.

Then, we show an application of this result to hypergraph sketching which is the problem of learning a sparse (both in the number of hyperedges and the size of the hyperedges) hypergraph from uniformly drawn random cuts. We also provide experimental results on a real world dataset.

## 1 Introduction

Learning sparse polynomials over the Boolean domain is one of the fundamental problems from computational learning theory and has been studied extensively over the last twenty-five years [1–6]. In almost all cases, known algorithms for learning or interpolating sparse polynomials require query access to the unknown polynomial. An outstanding open problem is to find an algorithm for learning $s$-sparse polynomials with respect to the uniform distribution on $\{-1,1\}^n$ that runs in time polynomial in $n$ and $g(s)$ (where $g$ is any fixed function independent of $n$) and requires only randomly chosen examples to succeed. In particular, such an algorithm would imply a breakthrough result for the problem of learning $k$-juntas (functions that depend on only $k \ll n$ input variables; it is not known how to learn $\omega(1)$-juntas in polynomial time).

We present an algorithm and a set of natural conditions such that any sparse polynomial $f$ satisfying these conditions can be learned from random examples in time polynomial in $n$ and $2^s$. In particular, any $f$ whose coefficients have been subjected to a small perturbation (smoothed analysis setting) satisfies these conditions (for example, if a Gaussian with arbitrarily small variance is added independently to each coefficient, $f$ satisfies these conditions with probability 1). We state our main result here:

**Theorem 1.** *Let $f$ be an $s$-sparse function that satisfies at least one of the following properties: a) (smoothed analysis setting)The coefficients $\{c_i\}_{i=1}^s$ are in general position or all of them are perturbed by a small random noise. b) The $s$ parity functions are linearly independent. c) All the coefficients are positive. Then we learn $f$ with high probability in time* $\mathrm{poly}(n, 2^s)$.

We note that smoothed-analysis, pioneered in [7], has now become a common alternative for problems that seem intractable in the worst-case.

Our algorithm also succeeds in the presence of noise:

**Theorem 2.** *Let $f = f_1 + f_2$ be a polynomial such that $f_1$ and $f_2$ depend on mutually disjoint set of parity functions. $f_1$ is s-sparse and the values of $f_1$ are 'well separated'. Further, $\|f_2\|_1 \leq \nu$, (i.e., f is approximately sparse). If observations are corrupted by additive noise bounded by $\epsilon$, then there exists an algorithm which takes $\epsilon + \nu$ as an input, that gives g in time polynomial in $n$ and $2^s$ such that $\|f - g\|_2 \leq O(\nu + \epsilon)$ with high probability.*

The treatment of the noisy case, i.e., the formal statement of this theorem, the corresponding algorithm, and the related proofs are relegated to the supplementary material. All these results are based on what we call as the *unique sign property:* If there is one value that $f$ takes which uniquely specifies the signs of the parity functions involved, then the function is efficiently learnable. Note that our results cannot be used for learning juntas or other Boolean-valued sparse polynomials, since the unique sign property does not hold in these settings.

We show that this property holds for the complement of the cut function on a hypergraph (no. of hyperedges $-$ cut value). This fact can be used to learn the cut complement function and eventually infer the structure of a sparse hypergraph from random cuts. Sparsity implies that the number of hyperedges and the size of each hyperedge is of constant size. Hypergraphs can be used to represent relations in many real world data sets. For example, one can represent the relation between the books and the readers (users) on the Amazon dataset with a hypergraph. Book titles and Amazon users can be mapped to nodes and hyperedges, respectively ([8]). Then a node belongs to a hyperedge, if the corresponding book is read by the user represented by that hyperedge. When such graphs evolve over time (and space), the difference graph filtered by time and space is often sparse. To locate and learn the few hyperedges from random cuts in such difference graphs constitutes hypergraph sketching. We test our algorithms on hypergraphs generated from the dataset that contain the time stamped record of messages between Yahoo! messenger users marked with the user locations (zip codes).

## 1.1 Approach and Related Work

The problem of recovering the sparsest solution of a set of underdetermined linear equations has received significant recent attention in the context of compressed sensing [9–11]. In compressed sensing, one tries to recover an unknown sparse vector using few linear observations (measurements), possibly in the presence of noise.

The recent papers [12,13] are of particular relevance to us since they establish a connection between learning sparse polynomials and compressed sensing. The authors show that the problem of learning a sparse polynomial is equivalent to recovering the unknown sparse coefficient vector using linear measurements. By applying techniques from compressed sensing theory, namely Restricted Isometry Property (see [12]) and incoherence (see [13]), the authors independently established results for reconstructing sparse polynomials using convex optimization. The results have near-optimal sample complexity. However, the running time of these algorithms is exponential in the underlying dimension, $n$. This is because the measurement matrix of the equivalent compressed sensing problem requires one column for every possible non-zero monomial.

In this paper, we show how to solve this problem in time polynomial in $n$ and $2^s$ under the assumption of *unique sign property* on the sparse polynomial. Our key contribution is a novel identification procedure that can reduce the list of potentially non-zero coefficients from the naive bound of $2^n$ to $2^s$ when the function has this property.

On the theoretical side, there has been interesting recent work of [14] that *approximately* learns sparse polynomial functions when the underlying domain is Gaussian. Their results do not seem to translate to the Boolean domain. We also note the work of [15] that gives an algorithm for learning sparse Boolean functions with respect to a *randomly* chosen product distribution on $\{-1, 1\}^n$. Their work does not apply to the uniform distribution on $\{-1, 1\}^n$.

On the practical side, we give an application of the theory to the problem of hypergraph sketching. We generalize a prior work [12] that applied the compressed sensing approach discussed before to

graph sketching on evolving social network graphs. In our algorithm, while the sample complexity requirements are higher, the time complexity is greatly reduced in comparison. We test our algorithms on a real dataset and show that the algorithm is able to scale well on sparse hypergraphs created out of Yahoo! messenger dataset by filtering through time and location stamps.

## 2 Definitions

Consider a real-valued function over the Boolean hypercube $f : \{-1, 1\}^n \to \mathbb{R}$. Given a sequence of labeled samples of the form $\langle f(\mathbf{x}), \mathbf{x} \rangle$, where $\mathbf{x}$ is sampled from the uniform distribution $U$ over the hypercube $\{-1, 1\}^n$, we are interested in an efficient algorithm that learns the function $f$ with high probability. Through Fourier expansion, $f$ can be written as a linear combination of monomials:

$$f(\mathbf{x}) = \sum_{S \subseteq [n]} c_S \chi_S(\mathbf{x}), \ \forall \, \mathbf{x} \in \{-1, 1\}^n \tag{1}$$

where $[n]$ is the set of integers from 1 to $n$, $\chi_S(\mathbf{x}) = \prod_{i \in S} x_i$ and $c_S \in \mathbb{R}$. Let $\mathbf{c}$ be the vector of coefficients $c_S$. A monomial $\chi_S(\mathbf{x})$ is also called a parity function. More background on Boolean functions and the Fourier expansion can be found in [16].

In this work, we restrict ourselves to *sparse polynomials* $f$ with sparsity $s$ in the Fourier domain, i.e., $f$ is a linear combination of unknown parity functions $\chi_{S_1}(\mathbf{x}), \chi_{S_2}(\mathbf{x}), \dots \chi_{S_s}(\mathbf{x})$ with $s$ unknown real coefficients given by $\{c_{S_i}\}_{i=1}^s$ such that $c_{S_i} \neq 0$, $\forall 1 \leq i \leq s$; all other coefficients are 0. Let the subsets corresponding to the $s$ parity functions form a family of sets $\mathcal{I} = \{S_i\}_{i=1}^s$. Finding $\mathcal{I}$ is equivalent to finding the $s$ parity functions.

**Note:** In certain places, where the context makes it clear, we slightly abuse the notation such that the set $S_i$ identifying a specific parity function is replaced by just the index $i$. The coefficients may be denoted simply by $c_i$ and the parity functions by $\chi_i(\cdot)$.

Let $\mathbb{F}_2$ denote the binary field. Every parity function $\chi_i(\cdot)$ can be represented by a vector $\mathbf{p}_i \in \mathbb{F}_2^{n \times 1}$. The $j$-th entry $\mathbf{p}_i(j)$ in the vector $\mathbf{p}_i$ is 1, if $j \in S_i$ and is 0 otherwise.

**Definition 1.** *A set of $s$ parity functions $\{\chi_i(\cdot)\}_{i=1}^s$ are said to be linearly independent if the corresponding set of vectors $\{\mathbf{p}_i\}_{i=1}^s$ are linearly independent over $\mathbb{F}_2$.*

Similarly, they are said to have rank $r$ if the dimension of the subspace spanned by $\{\mathbf{p}_i\}_{i=1}^s$ is $r$.

**Definition 2.** *The coefficients $\{c_i\}_{i=1}^s$ are said to be in general position if for all possible set of values $b_i \in \{0, 1, -1\}$, $\forall \, 1 \leq i \leq s$, with at least one nonzero $b_i$, $\sum_{i=1}^s c_i b_i \neq 0$*

**Definition 3.** *The coefficients $\{c_i\}_{i=1}^s$ are said to be $\mu$-separated if for all possible set of values $b_i \in \{0, 1, -1\}$, $\forall \, 1 \leq i \leq s$ with at least one nonzero $b_i$, $\left| \sum_{i=1}^s c_i b_i \right| > \mu$.*

**Definition 4.** *A sign pattern is a distinct vector of signs $\mathbf{a} = [\chi_1(\cdot), \chi_2(\cdot), \dots \chi_s(\cdot)] \in \{-1, 1\}^{1 \times s}$ assumed by the set of $s$ parity functions.*

Since this work involves switching representations between the real and the binary field, we define a function $q$ that does the switch.

**Definition 5.** *$q : \{-1, 1\}^{a \times b} \to \mathbb{F}_2^{a \times b}$ is a function that converts a sign matrix $\mathbf{X}$ to a matrix $\mathbf{Y}$ over $\mathbb{F}_2$ such that $Y_{ij} = q(X_{ij}) = 1 \in \mathbb{F}_2$, if $X_{ij} = -1$ and $Y_{ij} = q(X_{ij}) = 0 \in \mathbb{F}_2$, if $X_{ij} = 1$. Clearly, it has an inverse function $q^{-1}$ such that $q^{-1}(\mathbf{Y}) = \mathbf{X}$.*

We also present some definitions to deal with the case when the polynomial $f$ is not exactly $s$-sparse and observations are noisy. Let $2^{[n]}$ denote the power set of $[n]$.

**Definition 6.** *A polynomial $f : \{-1, 1\}^n \to \mathbb{R}$ is called approximately $(s, \nu)$-sparse if there exists $\mathcal{I} \subset 2^{[n]}$ with $|\mathcal{I}| = s$ such that $\sum_{S \in \mathcal{I}^c} |c_S| < \nu$, where $\{c_S\}$ are the Fourier coefficients as in (1).*

In other words, the sum of the absolute values of all the coefficients except the ones corresponding to $\mathcal{I}$ are rather small.

# 3 Problem Setting

Suppose $m$ labeled samples $\langle f(\mathbf{x}), \mathbf{x} \rangle_{i=1}^m$ are drawn from the uniform distribution $U$ on the Boolean hypercube. For any $\mathcal{B} \subseteq 2^{[n]}$, let $\mathbf{c}_\mathcal{B} \in \mathbb{R}^{2^n \times 1}$ be the vector of real coefficients such that $c_\mathcal{B}(S) = c_S$, $\forall S \in \mathcal{B}$ and $c_\mathcal{B}(S) = 0$, $\forall S \notin \mathcal{B}$. Let $\mathbf{A} \in \mathbb{R}^{m \times 2^n}$ be such that every row of $\mathbf{A}$ corresponds to one random input sample $\mathbf{x} \sim U$. Let $\mathbf{x}$ also denote the row index and $S \subseteq [n]$ denote the column index of $\mathbf{A}$. $\mathbf{A}(\mathbf{x}, S) = \chi_S(\mathbf{x})$. Let $A_\mathcal{S}$ denote the sub matrix formed by the columns corresponding to the subsets in $\mathcal{S}$. Let $\mathcal{I}$ be the set consisting of the $s$ parity functions of interest in both the sparse and the approximately sparse cases. A *sparse representation* of an approximately $(s, \nu)$-sparse function $f$ is $f_\mathcal{I} = \mathbf{A}(\mathbf{x}) \mathbf{c}_\mathcal{I}$, where $\mathbf{c}_\mathcal{I}$ is as defined above.

We review the compressed sensing framework used in [12] and [13]. Specifically, for the remainder of the paper, we rely on [13] as a point of reference. We review their framework and explain how we use it to obtain our results, particularly for the noisy case.

Let $\mathbf{y} \in \mathbb{R}^m$ and $\beta_\mathcal{S} \in \mathbb{R}^{2^n}$, such that $\beta_S = 0$, $\forall S \subseteq \mathcal{S}^c$. Note that, here $\mathcal{S}$ is a subset of the power set $2^{[n]}$. Now, consider the following convex program for noisy compressed sensing in this setting:

$$\min \|\beta_\mathcal{S}\|_1 \text{ subject to } \sqrt{\frac{1}{m}} \|\mathbf{A}\beta_\mathcal{S} - \mathbf{y}\|_2 \le \epsilon. \tag{2}$$

Let $\beta_\mathcal{S}^{\text{opt}}$ be an optimum for the program (2). Note that only the columns of $\mathbf{A}$ in $\mathcal{S}$ are used in the program. The convex program runs in time $\text{poly}(m, |\mathcal{S}|)$. The incoherence property of the matrix $A$ in [13] implies the following.

**Theorem 3.** *( [13]) For any family of subsets $\mathcal{I} \in 2^{[n]}$ such that $|\mathcal{I}| = s$, $m = 4096ns^2$ and $c_1 = 4, c_2 = 8$, for any feasible point $\beta_\mathcal{S}$ of program 2, we have:*

$$\|\beta_\mathcal{S} - \beta_\mathcal{S}^{\text{opt}}\|_2 \le c_1\epsilon + c_2 \left(\frac{n}{m}\right)^{1/4} \|\beta_{\mathcal{I}^c \cap \mathcal{S}}\|_1 \tag{3}$$

*with probability at least $1 - O\left(\frac{1}{4^n}\right)$*

When $\mathcal{S}$ is set to the power set $2^{[n]}, \epsilon = 0$ and $\mathbf{y}$ is the vector of observed values for an $s$-sparse polynomial, the $s$-sparse vector $\mathbf{c}_\mathcal{I}$ is a feasible point to program (2). By Theorem 3, the program recovers the sparse vector $\mathbf{c}_\mathcal{I}$ and hence learns the function. The only caveat is that the complexity is exponential in $n$.

The main idea behind our algorithms for noiseless and noisy sparse function learning is to 'capture' the actual $s$-sparse set $\mathcal{I}$ of interest in a small set $\mathcal{S} : |\mathcal{S}| = O(2^s)$ of coefficients by a separate algorithm that runs in time $\text{poly}(n, 2^s)$. Using the restricted set of coefficients $\mathcal{S}$, we search for the sparse solution under the noisy and noiseless cases using program (2).

**Lemma 1.** *Given an algorithm that runs in time $\text{poly}(n, 2^s)$ and generates a set of parities $\mathcal{S}$ such that $|\mathcal{S}| = O(2^s), \mathcal{I} \subseteq \mathcal{S}$ with $|\mathcal{I}| = s$, program (2) with $\mathcal{S}$ and $m = 4096ns^2$ random samples as inputs runs in time $\text{poly}(n, 2^s)$ and learns the correct function with probability $1 - O\left(\frac{1}{4^n}\right)$.*

**Unique Sign Pattern Property:** The key property that lets us find a small $\mathcal{S}$ efficiently is the unique sign pattern property. Observe that an $s$-sparse function can produce at most $2^s$ different real values. If the maximum value obtained always corresponds to a *unique pattern of signs of parities*, by looking only at the random samples $\mathbf{x}$ corresponding to the subsequent $O(n)$ occurrences of this maximum value, we show that all the parity functions needed to learn $f$ are captured in a small set of size $2^{s+1}$ (see Lemma 2 and its proof). The unique sign property again plays an important role, along with Theorem 3 with more technicalities added, in the noisy case, which we visit in Section 2 of the supplementary material.

In the next section, we provide an algorithm to generate the bounded set $\mathcal{S}$ for the noiseless case for an $s$-sparse function $f$ and provide guarantees for the algorithm formally.

# 4 Algorithm and Guarantees: Noiseless case

Let $\mathcal{I}$ be the family of $s$ subsets $\{S_i\}_{i=1}^s$ each corresponding to the $s$ parity functions $\chi_{S_i}(\cdot)$ in an $s$-sparse function $f$. In this section, we provide an algorithm, named *LearnBool*, that finds a small

subset $\mathcal{S}$ of the power set $2^{[n]}$ that contains elements of $\mathcal{I}$ first and then uses program (2) with $\mathcal{S}$. We show that the algorithm learns $f$ in time $\mathrm{poly}\,(n, 2^s)$ from uniformly randomly drawn labeled samples from the Boolean hypercube with high probability under some natural conditions.

Recall that if the function is such that $f(\mathbf{x})$ attains its maximum value only if $[\chi_1(\mathbf{x}), \chi_2(\mathbf{x}) \ldots \chi_s(\mathbf{x})] = \mathbf{a}_{max} \in \{-1, 1\}^s$ for some unique sign pattern $\mathbf{a}_{\max}$, then the function is said to possess the *unique sign property*. Now we state the main technical lemma for the unique sign property.

**Lemma 2.** *If an $s$-sparse function $f$ has the unique sign property then, in Algorithm 1, $\mathcal{S}$ is such that $\mathcal{I} \subseteq \mathcal{S}$, $|\mathcal{S}| \leq 2^{s+1}$ with probability $1 - O\left(\frac{1}{n}\right)$ and runs in time $\mathrm{poly}(n, 2^s)$.*

*Proof.* See the supplementary material. □

The proof of the above lemma involves showing that the random matrix $\mathbf{Y}_{\max}$ (see Algorithm 1) has rank at least $n - s$, leading to at most $2^s$ solutions for each equation in (4). The feasible solutions can be obtained by Gaussian elimination in the binary field.

**Theorem 4.** *Let $f$ be an $s$-sparse function that satisfies at least one of the following properties:*

*(a) The coefficients $\{c_i\}_{i=1}^s$ are in general position.*
*(b) The $s$ parity functions are linearly independent.*
*(c) All the coefficients are positive.*

*Given labeled samples, Algorithm 1 learns $f$ exactly (or $\mathbf{v}^{\mathrm{opt}} = \mathbf{c}$) in time $\mathrm{poly}\,(n, 2^s)$ with probability $1 - O\left(\frac{1}{n}\right)$.*

*Proof.* See the supplementary material. □

*Smoothed Analysis Setting:* Perturbing $c_i$'s with Gaussian random variables of standard deviation $\sigma > 0$ or by random variables drawn from any set of reasonable continuous distributions ensures that the perturbed function satisfies property (a) with probability 1.

*Random Parity Functions:* When $c_i$'s are arbitrary and the set of $s$ parity functions are drawn uniformly randomly from $2^{[n]}$, then property (b) holds with high probability if $s$ is a constant.

---

**Input**: Sparsity parameter $s$, $m_1 = 2n2^s$ random labeled samples $\{\langle f(\mathbf{x}_i), \mathbf{x}_i \rangle\}_{i=1}^{m_1}$.
Pick samples $\{\mathbf{x}_{i_j}\}_{j=1}^{n_{\max}}$ corresponding to the maximum value of $f$ observed in all the $m$ samples.
Stack all $\mathbf{x}_{i_j}$ row wise into a matrix $\mathbf{X}_{\max}$ of dimensions $n_{\max} \times n$.
Initialise $\mathcal{S} = \emptyset$. Let $\mathbf{Y}_{\max} = q(\mathbf{X}_{\max})$.
Find all feasible solutions $\mathbf{p} \in \mathbb{F}_2^{n \times 1}$ such that:

$$\mathbf{1}_{n_{\max} \times 1} = \mathbf{Y}_{\max} \mathbf{p} \text{ or } \mathbf{0}_{n_{\max} \times 1} = \mathbf{Y}_{\max} \mathbf{p} \tag{4}$$

Collect all feasible solutions $\mathbf{p}$ to either of the above equations in the set $P \subseteq \mathbb{F}_2^{n \times 1}$.
$\mathcal{S} = \{\{j \in [n] : \mathbf{p}(j) = 1\} | \mathbf{p} \in P\}$.
Using $m = 4096ns^2$ more samples (number of rows of $\mathbf{A}$ is $m$ corresponding to these new samples), solve:

$$\beta_{\mathcal{S}}^{\mathrm{opt}} = \min\|\beta_{\mathcal{S}}\|_1 \text{ such that } \mathbf{A}\beta_{\mathcal{S}} = \mathbf{y}, \tag{5}$$

where $\mathbf{y}$ is the vector of $m$ observed values.
Set $\mathbf{v}^{\mathrm{opt}} = \beta_{\mathcal{S}}^{opt}$.
**Output**: $\mathbf{v}^{\mathrm{opt}}$.

**Algorithm 1:** LearnBool

---

## 5    A Sparse Polynomial Learning Application: Hypergraph Sketching

Hypergraphs can be used to model the relations in real world data sets (e.g., books read by users in Amazon). We show that the cut functions on hypergraphs satisfy the unique sign property. Learning a cut function of a sparse hypergraph from random cuts is a special case of learning a sparse

polynomial from samples drawn uniformly from the Boolean hypercube. To track the evolution of large hypergraphs over a small time interval, it is enough to learn the cut function of the difference graph which is often sparse. This is called the *graph sketching problem*. Previously, graph sketching was applied to social network evolution [12]. We generalize this to hypergraphs showing that they satisfy the unique sign property, which enable faster algorithms, and provide experimental results on real data sets.

## 5.1 Graph Sketching

A *hypergraph* $G = (V, E)$ is a set of vertices $V$ along with a set $E$ of subsets of $V$ called the *hyperedges. The size of a hyperedge* is the number of variables that the hyperedge connects. Let $d$ be the maximum hyperedge size of graph $G$. Let $|V| = n$ and $|E| = s$.

A random cut $S \subseteq V$ is a set of vertices selected uniformly at random. Define the value of the cut $S$ to be $c(S) = |\{e \in E : e \cap S \neq \emptyset, \ e \cap V - S \neq \emptyset\}|$. Graph sketching is the problem of identifying the graph structure from random queries that evaluate the value of a random cut, where $s \ll n$ (sparse setting). Hypergraphs naturally specify relations among a set of objects through hyperedges. For example, Amazon users can form the set $E$ and Amazon books can form the set $V$. Each user may read a subset of books which represents the hyperedge. Learning the hypergraph corresponds to identifying the sets of books bought by each user. For more examples of hypergraphs in real data sets, we refer the reader to [8]. Such hypergraphs evolve over time. The difference graph between two consecutive time instants is expected to be sparse (number of edges $s$ and maximum hyperedge size $d$ are small). We are interested in learning such hypergraphs from random cut−queries.

For simplicity and convenience, we consider *the cut complement* query, i.e., c−cut, which returns $s - c(S)$. One can easily represent the c−cut query with a sparse polynomial as follows: Let node $i$ correspond to variable $x_i \in \{-1, +1\}$. A random cut involves choosing $x_i$ uniformly randomly from $\{-1, +1\}$. The variables assigned to $+1$ belong to the random cut $S$. The value is given by the polynomial

$$f_{c-cut}(\mathbf{x}) = \sum_{\mathcal{I} \in E} \left( \prod_{i \in \mathcal{I}} \frac{(1 + x_i)}{2} + \prod_{i \in \mathcal{I}} \frac{(1 - x_i)}{2} \right) = \sum_{\mathcal{I} \in E} \frac{1}{2^{|\mathcal{I}|-1}} \left( \sum_{\substack{\mathcal{J} \subseteq \mathcal{I}, \\ |\mathcal{J}| \text{ is even}}} \left( 1 + \prod_{i \in \mathcal{J}} x_i \right) \right). \quad (6)$$

Hence, the c−cut function is a sparse polynomial where the sparsity is at most $s2^{d-1}$. The variables corresponding to the nodes that belong to some hyperedge appear in the polynomial. We call these *the relevant variables* and the number of relevant variables is denoted by $k$. Note that, in our sparse setting $k \leq sd$. We note that for a hypergraph with no singleton hyperedge, given the c−cut function, it is easy to recover the hyper edges from (6). Therefore, we focus on learning the c−cut function to sketch the hypergraph.

When $G$ is a graph with edges (of cardinality 2), the compressed sensing approach (using program 2) using the cut (or c−cut) values as measurements is shown to be very efficient in [12] in terms of the sample complexity, i.e., the required number of queries. The run time is efficient because total number of candidate parities is $O(n^2)$. However when we consider hypergraphs, i.e., when $d$ is a large constant, the compressed sensing approach cannot scale computationally ($\text{poly}(n^d)$ runtime). Here, based on the theory developed, we give a faster algorithm based on the unique sign property with sample complexity $m_1 = O(2^k d \log n + 2^{2d+1} s^2 (\log n + k))$ and run time of $O(m_1 2^k, n^2 \log n))$.

We observe that the c−cut polynomial satisfies the unique sign property. From (6), it is evident that the polynomial has only positive coefficients. Therefore, by Theorem 4, algorithm LearnBool succeeds. The maximum value of the c−cut function is the number of edges. Notice that the maximum value is definitely observed in two configurations of the relevant variables: If either all relevant variables are $+1$ or all are $-1$. Therefore, the maximum value is observed in every $2^{k-1} \leq 2^{sd}$ samples. Thus, a direct application of *LearnBool* yields $\text{poly}(n, 2^{k-1})$ time complexity, which improves the $O(n^d)$ bound for small $s$ and $d$.

Improving further, we provide a more efficient algorithm tailored for the hypergraph sketching problem, which makes use of the unique sign property and some other properties of the cut function. Algorithm LearnGraph (Algorithm 4) is provided in the supplementary material.

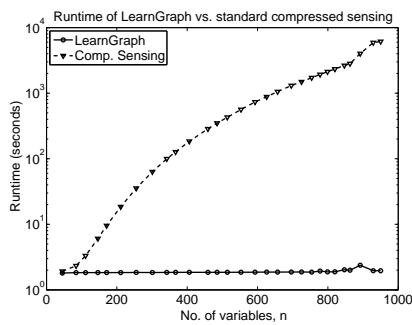
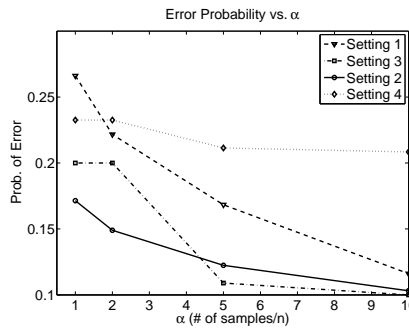

(a) Runtime vs. # of variables, $d = 3$ and $s = 1$.

(b) Probability of error vs. $\alpha$.

Figure 1: Performance figures comparing LearnGraph and Compressed Sensing approach.

**Theorem 5.** *Algorithm 4 exactly learns the $c-$cut function with probability $1 - O(\frac{1}{n})$ with sample complexity $m_1 = O(2^k d \log n + 2^{2d+1} s^2 (\log n + k))$ and time complexity $O(2^k m_1 + n^2 d \log n))$.*

*Proof.* See the supplementary material. □

### 5.2 Yahoo! Messenger User Communication Pattern Dataset

We performed simulations using MATLAB on an Intel(R) Xeon(R) quad-core 3.6 GHz machine with 16 GB RAM and 10M cache. We run our algorithm on the Yahoo! Messenger User Communication Pattern Dataset [17]. This dataset contains the timestamped user communication data, i.e., information about a large number of messages sent over Yahoo! Messenger, for a duration of 28 days.

*Dataset:* Each row represents a message. The first two columns show the day and time (time stamp) of the message respectively. The third and fifth columns show the ID of the transmitting and receiving users, respectively. The fourth column shows the zipcode (spatial stamp) from which this particular message is transmitted. The sixth column shows if the transmitter was in the contact list of the reciver user (y) or not (n). If a transmitter sends the same receiver more than one message from the same zipcode, only the first message is shown in the dataset. In total, there are 100000 unique users and 5649 unique zipcodes.

We form a hypergraph from the dataset as follows: The transmitting users form the hyperedges and the receiving users form the nodes of the hypergraph. A hyperedge connects a set $T$ of users if there is a transmitting user that sends a message to all the users in $T$. In any given time interval $\delta t$ (short time interval) and small set of locations $\delta x$ specified by the number of zip codes, there are few users who transmit ($s$) and they transmit to very few users ($d$). The complete set of nodes in the hypergraph ($n$) is taken to be those receiving users who are active during $m$ consecutive intervals of length $\delta t$ and in a set of $\delta x$ zipcodes. This gives rise to a sparse graph. We identify the active set of transmitting users (hyperedges) and their corresponding receivers (nodes in these hyperedges) during a short time interval $\delta t$ and a randomly selected space interval ($\delta x$, i.e., zip codes) from a large pool of receivers (nodes) that are observed during $m$ intervals of length $\delta t$. Details of $\delta t$, $m$ and $\delta x$ chosen for experiments are given in Table 1. We note that $n$ is in the order of 1000 usually.

**Remark:** Our task is to learn the $c-$cut function from the random queries, i.e., random +/-1 assignment of variables and corresponding $c-$cut values. The generated sparse graph contains only hyperedges that have more than 1 node. Other hyperedges (transmitting users) with just one node in the sparse hypergraph are not taken into account. This is because a singleton hyperedge $i$ is always counted in the $c-$cut function thereby effectively its presence is masked. First, we identify the relevant variables that participate in the sparse graph. After identifying this set of candidates, correlating the corresponding candidate parities with the function output yields the Fourier coefficient of that parity (see Algorithm 4).

Table 1: Runtime for different graphs. **LG:** LearnGraph, **CS:** Compressed sensing based alg.

(a) Runtime for $d = 4$ and $s = 1$ graph.

| $n$ / Alg. | 88 | 159 | 288 | 556 | 1221 |
|---|---|---|---|---|---|
| LG | 1.96 | 2.13 | 2.23 | 2.79 | 4.94 |
| CS | 265.63 | - | - | - | - |

(b) Runtime for $d = 4$ and $s = 3$ graph.

| $n$ / Alg. | 52 | 104 | 246 | 412 | 1399 |
|---|---|---|---|---|---|
| LG | 1.91 | 2.08 | 2.08 | 2.30 | 4.98 |
| CS | 39.89 | > 10823 | - | - | - |

(c) Simulation parameters for Fig. 1b

| Setting No. | Interval | # of Int. | $n$ | $\max(d)$ | $\max(s)$ | Zip. Set Size |
|---|---|---|---|---|---|---|
| Setting 1 | 5 min. | 20 | 6822 | 10 | 19 | 20 |
| Setting 2 | 20 sec. | 200 | 5730 | 22 | 4 | 200 |
| Setting 3 | 10 min. | 10 | 6822 | 11 | 13 | 2 |
| Setting 4 | 2 min. | 50 | 6822 | 30 | 21 | 50 |

### 5.2.1 Performance Comparison with Compressed Sensing Approach

First, we compare the runtime of our implementation $\mathrm{LearnGraph}$ with the compressed sensing based algorithm from [12]. Both algorithms correctly identify the relevant variables in all the considered range of parameters. The last step of finding the corresponding Fourier coefficients is omitted and can be easily implemented (Algorithm 4) without significantly affecting the running time. As can be seen in Tables 1a, 1b and Fig. 1a, LearnGraph scales well to graphs on thousands of nodes. On the contrary, the compressed sensing approach must handle a measurement matrix of size $O(n^d)$, which becomes prohibitively large on graphs involving more than a few hundred nodes.

### 5.2.2 Error Performance of LearnGraph

Error probability (probability that the correct $c-$cut function is not recovered) versus the number of samples used is plotted for four different experimental settings of $\delta t$, $\delta x$ and $m$ in Fig. 1b. For each time interval, the error probability is calculated by averaging the number of errors among 100 different trials. For each value of $\alpha$ (number of samples), the error probability is averaged over time intervals to illustrate the error performance. We only keep the intervals for which the graph filtered with the considered zipcodes contains at least one user with more than one neighbor. We find that for the first 3 settings, the error probability decreases with more samples. For the fourth setting, $d$ and $s$ are very large and hence a large number of samples are required. For that reason, the error probability does not improve significantly. The probability of error can be reduced by repeating the experiment multiple times and taking a majority, at the cost of significantly more samples. Our plot shows only the probability of error without such a majority amplification.

## 6 Conclusions

We presented a novel algorithm for learning sparse polynomials by random samples on the Boolean hypercube. While the general problem of learning all sparse polynomials is notoriously hard, we show that almost all sparse polynomials can be efficiently learned using our algorithm. This is because our unique sign property holds for randomly perturbed coefficients, in addition to several other natural settings. As an application, we show that graph and hypergraph sketching lead to sparse polynomial learning problems that always satisfy the unique sign property. This allows us to obtain efficient reconstruction algorthms that outperform the previous state of the art for these problems.

An important open problem is to achieve the sample complexity of [12] while keeping the computational complexity polynomial in $n$.

### Acknowledgments

M.K, K.S. and A.D. acknowledge the support of NSF via CCF 1422549, 1344364, 1344179 and DARPA STTR and a ARO YIP award.

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
