[Supplementary Material · appendices.pdf]

# Appendix for "Sparse Polynomial Learning and Graph Sketching"

**Murat Kocaoglu**[1*], **Karthikeyan Shanmugam**[1†], **Alexandros G.Dimakis**[1‡], **Adam Klivans**[2⋆]
[1]Department of Electrical and Computer Engineering, [2]Department of Computer Science
The University of Texas at Austin, USA
*mkocaoglu@utexas.edu, †karthiksh@utexas.edu
‡dimakis@austin.utexas.edu, ⋆klivans@cs.utexas.edu

## 1  Proof of Theorem 4

We prove Theorem 4 at the end of this section. Next, we provide the proof for Lemma 2 about Algorithm 1 that will be used in the proof. Since the function $f$ is $s$-sparse, it takes at most $2^s$ distinct real values.

**Proof of Lemma 2**

Let $E_1$ be the event that the maximum value observed among $m_1$ samples in the algorithm 1 is the maximum value attained by $f$. Note that, the probability that the function attains the maximum value is at least $\frac{1}{2^s}$. To see this, if the parity functions have rank $r$, then the set of $r$ linearly independent parity functions take values uniformly in the hypercube $\{-1, 1\}^r$ and other are determined by these $r$ signs. Hence, the probability of finding the maximum value is $\frac{1}{2^r} \geq \frac{1}{2^s}$. If the functions satisfies the unique sign property for the maximum value and if $E_1$ is true, it is easily seen that the actual party functions $\mathbf{p}_i$ are in the set $P$ in the algorithm 1.

Consider the algorithm 1. Let $E_3$ be the event that the matrix $\mathbf{Y}_{\max}$ has at least rank $n - s$. $E_3$ implies that $|P| = |\mathcal{S}| \leq 2^{s+1}$, $\forall 1 \leq i \leq s$. Let $E_2$ be the event that $n_{\max} > 2n$. Conditioned on $E_2$ and $E_1$ being true, we first argue that the rank of $\mathbf{Y}_{\max}$ is at least $n - s$ with high probability. Let the rank of the actual set of parity functions $[\mathbf{p}_1, \mathbf{p}_2 \dots \mathbf{p}_s]$ be $k \leq s$.

If $E_1$ and $E_2$ are true, then $\mathbf{Y}_{\max}$ contains $2n$ random samples such that they all produce the same sign pattern $\mathbf{a}_{\max}$ because the actual function $f$ satisfies the unique sign pattern property for the maximum value. Let $\mathbf{z}_{\max} = q(\mathbf{a}_{\max})$. Observe that rows of $\mathbf{Y}_{\max}$ are random samples uniformly drawn from the hyperplane $\mathcal{H} = \{\mathbf{x} \in \mathbb{F}_2^{n \times 1} : \mathbf{x}^T[\mathbf{p}_i] = z_{\max}(i), \forall 1 \leq i \leq s\}$. Since the rank of the parity functions is $k$, the dimension of $\mathcal{H}$ is $n - k$. Now, the rank of space spanned by $2n$ samples drawn randomly uniformly from $\mathcal{H}$ is at least the rank of space spanned by $2n$ samples drawn randomly uniformly from $\mathbb{F}_2^{1 \times n-k}$. The probability that a random $2n \times n - k$ binary matrix is full rank is given by:

$$\Pr(\text{a random } 2n \times n - k \text{ binary matrix is full rank}) = \prod_{i=0}^{n-k-1}\left(1 - \frac{1}{2^{2n-i}}\right) \geq \left(1 - \frac{1}{2^n}\right)^{n-k}$$

$$\geq \left(1 - \frac{1}{2^n}\right)^n \geq 1 - O\left(\frac{1}{n}\right) \tag{1}$$

Hence, $\Pr(E_3|E_2 \bigcap E_1) \geq 1 - O\left(\frac{1}{n}\right)$. $\Pr(E_1 \bigcap E_2)$ is the probability that there are at least $n_{\max}$ samples corresponding to the maximum value of the actual function in the $2n2^s$ samples drawn. Therefore, $\Pr(E_1 \bigcap E_2) \geq 1 - \left(1 - \frac{1}{2^s}\right)^{2n2^s - 2n}$ because the maximum value of $f$ is seen with

probability at least $\frac{1}{2^s}$. Using this in the following chain, we have:

$$\Pr\left(|\mathcal{S}| \leq 2^{s+1}, \mathcal{I} \subseteq \mathcal{S}\right) \geq \Pr\left(E_1 \bigcap E_2 \bigcap E_3\right) \geq \Pr\left(E_1 \bigcap E_2\right) \Pr\left(E_3 | E_2 \bigcap E_1\right)$$

$$\geq \Pr\left(E_1 \bigcap E_2\right)\left(1 - O\left(\frac{1}{n}\right)\right) \quad \text{(by (1))}$$

$$\geq \left(1 - \left(1 - \frac{1}{2^s}\right)^{2n2^s - 2n}\right)\left(1 - O\left(\frac{1}{n}\right)\right)$$

$$\geq \left(1 - \exp\left(-2n\left(1 - \frac{1}{2^s}\right)\right)\right)\left(1 - O\left(\frac{1}{n}\right)\right)$$

$$\geq \left(1 - O\left(\frac{1}{n}\right)\right)\left(1 - O\left(\frac{1}{n}\right)\right) \geq 1 - O\left(\frac{1}{n}\right). \tag{2}$$

Now, we relate the unique sign property to the conditions mentioned in Theorem 4 for its proof.

**Proof of Theorem 4**

Due to Lemmas 2 and 1, we just need to show that each of the conditions in the theorem implies the unique sign property, i.e., the maximum value of the function $f$ is attained when the set of parity functions takes a unique sign pattern.

*Case 1:* If the coefficients are in general position (Definition 2), all values taken by the function correspond to distinct sign patterns. This implies the unique sign property for the maximum value.

*Case 2:* If all the parity functions are linearly independent, any sign pattern can be realized. Then, the sign pattern $[\text{sign}\,(c_1)\,,\text{sign}\,(c_2)\ldots\text{sign}\,(c_s)]$ can be realized by the set of parity functions and this produces the value $\sum_{i=1}^{s}|c_i|$. And any other sign pattern will produce a strictly lesser value as all $c_i$ are nonzero. Hence, the maximum value is unique in this case.

*Case 3:* Let us consider the case when all the coefficients are positive. Even if the parity functions are linearly dependent, the sign pattern with all $+1$'s can be produced and this attains the unique maximum value $\sum_{i=1}^{s}|c_i|$. This implies the unique sign property.

## 2   Algorithms and Guarantees: Noisy Case

In this section, we provide our algorithm for learning an approximately $(s, \nu)$-sparse function with noisy samples, and prove guarantees regarding the error between the function learnt and the actual function. When $m$ random samples are observed, the noisy output model for an approximately $(s, \nu)$-sparse function $f$ is given by:

$$\mathbf{y} = \mathbf{Ac} + \varepsilon \tag{3}$$

where $\mathbf{A}$ is the $m$ by $2^n$ matrix where each row corresponds to a sample $\mathbf{x}$ and each column corresponds to a parity function and $\mathbf{c}$ is the set of Fourier coefficients for $f$ and the noise $|\varepsilon_i| \leq \epsilon$, $1 \leq i \leq m$. We recall that $\sum_{S \subseteq \mathcal{I}^c} |c_S| < \nu$ for an approximately sparse $f$. We assume that $\epsilon + \nu$ is known.

---

**Input**: The sequence of labeled samples $\langle f(\mathbf{x}_i) + \varepsilon_i, \mathbf{x}_i \rangle_{i=1}^{m}$
Initialise $\mathbf{X}_{max} = \emptyset$.
Let $\eta$ be the maximum value observed.
Stack all the inputs $\mathbf{x}_i$, such that $f(\mathbf{x}_i) + \varepsilon_i$ is in the neighborhood of radius $2(\epsilon + \nu)$ around $\eta$, into $\mathbf{X}_{max}$.
**Output**: $\mathbf{X}_{max}$.

**Algorithm 2:** MaxCluster

---

**Algorithm 3:** LearnBoolNoisy

Now we state our main thoerem for learning a sparse function from noisy observations.

**Theorem 6.** *Assume $f$ is an approximately $(s, \nu)$-sparse function as given in Definition 6 and observed samples satisfy the noise model in (3). Then, Algorithm 3 outputs $\mathbf{v}^{\mathrm{opt}}$ in time $\mathrm{poly}(n, 2^s)$ with probability $1 - O\left(\frac{1}{n}\right)$ satisfying $\|\mathbf{c} - \mathbf{v}^{\mathrm{opt}}\|_2 \leq \alpha_1\epsilon + \alpha_2\nu$, if $f$ satisfies at least one of the following properties:*

*(a) The coefficients $\{c_S\}_{S \in \mathcal{I}}$ are $4(\nu + \epsilon)$-separated.*
*(b) The set of parity functions $\chi_i(\cdot)$ are linearly independent, and $\min_{S \in \mathcal{I}} c_S > 4(\epsilon + \nu)$.*
*(c) All the coefficients are positive, and $\min_{S \in \mathcal{I}} c_S > 4(\epsilon + \nu)$.*

*Here, $\alpha_1$ and $\alpha_2$ are some constants.*

# 3 Proof of Theorem 6

Although the observations are noisy as in the noise model given by (3), the set of inputs for which the sparse representation of the function $f$, i.e., $f_{\mathcal{I}}$ (this depends on only Fourier coefficients in $\mathcal{I}$) attains its maximum, can still be perfectly identified under certain conditions given in the Lemma below. Algorithm 2 identifies those inputs.

**Lemma 3.** *If the function $f$ is approximately $(s, \nu)$-sparse, observations follow the noise model in (3), and if the values of $f_{\mathcal{I}}$ are separated by at least $4(\epsilon + \nu)$, then the output matrix $\mathbf{X}_{\max}$ in Algorithm 2 will contain exactly those inputs for which $f_{\mathcal{I}}$ attains the maximum value among the drawn samples.*

*Proof.* Consider a sample $\mathbf{x}$. Clearly, from the noise model and the definition of approximate sparsity, $|f(\mathbf{x}_i + \varepsilon_i) - f_{\mathcal{I}}(\mathbf{x}_i)| \leq \nu + \epsilon$. Hence, when using a radius of $2(\nu + \epsilon)$ for clustering, clearly no two samples with different $f_{\mathcal{I}}$ will be included in $\mathbf{X}_{max}$ and definitely one sample belonging to the maximum $f_{\mathcal{I}}$ among the observed samples will be included. $\square$

*Proof of Theorem 6:*

The three properties in the statement of Theorem 6 imply that $f_{\mathcal{I}}$ has the unique sign property for the maximum value due to the same arguments in the proof of Theorem 4. Further, they also imply that the values of $f_{\mathcal{I}}$ are separated by $4(\epsilon + \nu)$ in each of the cases. By Lemma 3, rows of $\mathbf{X}_{\max}$ contain only the inputs at which $f_{\mathcal{I}}$ attains its maximum among the observed values.

Using Lemma 2 on $f_{\mathcal{I}}$, which is exactly $s$-sparse, it can be seen that $|\mathcal{S}| \leq 2^{s+1}$ and contains all the parity functions in $f_{\mathcal{I}}$ with probability $1 - O\left(\frac{1}{n}\right)$ as in Algorithm 1. This is because $P$ is formed using inputs in $\mathbf{X}_{\max}$ that give the maximum $f_{\mathcal{I}}$ among the observed samples in an identical fashion

as in Algorithm 1. Now, we have the following chain of inequalities:

$$
\begin{aligned}
\|\mathbf{c} - \mathbf{v}^{\mathrm{opt}}\|_2 &\leq \|\mathbf{c}_\mathcal{S} - \beta_\mathcal{S}^{\mathrm{opt}}\|_2 + \|\mathbf{c}_{\mathcal{S}^c}\|_2 && \text{(triangle inequality)} \\
&\leq \|\mathbf{c}_\mathcal{S} - \beta_\mathcal{S}^{\mathrm{opt}}\|_2 + \|\mathbf{c}_{\mathcal{S}^c}\|_1 && (\|\cdot\|_2 \leq \|\cdot\|_1) \\
&\overset{a}{\leq} c_1(\nu + \epsilon) + c_2 \left(\frac{n}{m}\right)^{1/4} \|\mathbf{c}_{\mathcal{I}^c \cap \mathcal{S}}\|_1 + \|\mathbf{c}_{\mathcal{S}^c}\|_1 \\
&\leq c_1(\nu + \epsilon) + c_2(\nu) + \nu && (n < m)
\end{aligned}
\tag{5}
$$

For inequality (a), it is easy to see that $\mathbf{c}_\mathcal{S}$ is a feasible solution to program 4 and therefore Theorem 3 can be applied with $\beta_\mathcal{S} = \mathbf{c}_\mathcal{S}$ with noise threshold $\nu + \epsilon$. Further, $\|\mathbf{c}_{\mathcal{I}^c}\|_1 < \nu$.

Since $|\mathcal{S}| \leq 2^{s+1}$, the optimization program 4 runs in time poly $(n, 2^s)$.

## 4 Algorithm LearnGraph

We provide the algorithm LearnGraph below. Let $k$ be the number of relevant variables, i.e. variables that are part of at least one hyperedge. Note that $k \leq sd$.

---

**Input**: Number of edges $s$, $m_1 = \max c 2^k d \log n, c 2^{2d+1} s^2 (\log n + k)$ random labeled samples $\{\langle f_{c-cut}(\mathbf{x}_i), \mathbf{x}_i\rangle\}_{i=1}^{m_1}$.
Pick samples $\{\mathbf{x}_{i_j}\}_{j=1}^{n_{\max}}$ corresponding to the maximum value of $f_{c-cut}$ observed in all the $m$ samples. Stack all $\mathbf{x}_{i_j}$ row wise into a matrix $\mathbf{X}_{\max}$ of dimensions $n_{\max} \times n$.
$\mathbf{R} \Leftarrow \mathbf{X}_{\max}^T \mathbf{X}_{\max}$.
Estimate $d$ by $d = \max_i |\{j : \mathbf{R}(i,j) = \max(\mathbf{R}(i,:))\}|$
$c_0 = \dfrac{\sum\limits_{i=1}^{m_1} f_{c-cut}(\mathbf{x}_i)}{m_1}$
Identify the constant Fourier coefficient by rounding $c_0$ to the nearest integer multiple of $\frac{1}{2^d}$,
$c_0 \Leftarrow \dfrac{\mathrm{round}(c_0 2^d)}{2^d}$.
$f_{c-cut} \Leftarrow f_{c-cut} - c_0$.
Initialize $\mathcal{S}_i = \emptyset \; \forall i \in \{1, 2, ..., n\}$
Stack $x_j$ for all $(i, j)$ s.t. $\mathbf{R}(i, j) = n_{\max}$, into $\mathcal{S}_i$.
For all $M_{k_i} \subseteq \mathcal{S}_i$ s.t. $|M_{k_i}| \leq d$, $|M_{k_i}|$ is even, calculate $c_{\chi_{M_{k_i}}} = \dfrac{\sum\limits_{i=1}^{m_1} f_{c-cut}(\mathbf{x}_i)\chi_{M_{k_i}}(\mathbf{x}_i)}{m_1}$.
Find Fourier coefficients of parities by rounding $\chi_\mathcal{M}$ to the nearest integer multiple of $\frac{1}{2^d}$,
$c_{\chi_\mathcal{M}} \Leftarrow \dfrac{\mathrm{round}(c_{\chi_\mathcal{M}} 2^d)}{2^d}$
Stack all non-zero parity coefficients and parity variables into $\mathbf{c}$ and $\mathcal{M}$, respectively.
**Output**: $\mathbf{c}, \mathcal{M}$.

**Algorithm 4:** LearnGraph

---

**Note:** In the above algorithm, $\mathrm{round}(.)$ function rounds a real number to the nearest integer.

**Lemma 4.** *(Chernoff's bound) [1] Let $X_i$, $1 \leq i \leq n$ be i.i.d random variables taking values in $[b, c]$. Let $X = \sum\limits_{i=1}^n X_i$. Let $\mathbb{E}[X] = \mu$. Then,* $\Pr\left(\sum\limits_i X_i \geq \mu + a\right) \leq \exp\left(-\frac{a^2}{2(b-c)^2 n}\right)$ *and* $\Pr\left(\sum\limits_i X_i \leq \mu - a\right) \leq \exp\left(-\frac{a^2}{2(b-c)^2 n}\right)$.

*Proof of Theorem 5:*

Without loss of generality, let us consider the case when a hyperedge involves more than two vertices. Let us consider a variable to be relevant only if it is involved in at least one hyperedge with more than one vertex. Note that the number of relevant variables is $k \leq sd$. The c$-$cut function counts a hyperedge if either all its nodes are assigned $+1$ or when all its nodes are assigned $-1$. When the c$-$cut function attains its maximum values, every hyperedge is counted. Clearly, when all the relevant variables are assigned the same value from $\{+1, -1\}$, then c$-$cut attains its maximum value. This happens with probability $1/2^{k-1}$. Let $n_1$ denote the number of samples where all

relevant variables are assigned the same sign. Therefore, out of $m_1$ samples taken,

$$\Pr\left(n_1 \geq cd\log n\right) \geq 1 - \left(1 - \frac{1}{2^{k-1}}\right)^{m_1 - cd\log n}$$

$$\geq 1 - \exp\left(-2\left(1 - \frac{1}{2^k}\right)cd\log n\right)$$

$$\geq 1 - O\left(\frac{1}{n^{cd}}\right) \tag{6}$$

Therefore, $n_{\max} \geq cd\log n$ with very high probability. Let $E_1$ denote the event $n_{\max} \geq cd\log n$. Suppose $E_1$ is true, then any two variables that belong to the same hyperedge will have identical columns in $\mathbf{X}_{max}$. Therefore, if $i$ and $j$ are in the same hyperedge , then $R(i,j) = n_{\max}$. Let $\hat{\mathbf{x}}_i$ be the $i$-th column consisting of signs of the $i$-th variable. Let $x_{ik}$ be the $k$-th entry of the $i$th column. Then, $R(i,j) = \hat{\mathbf{x}}_i^T \hat{\mathbf{x}}_j = \sum_{k=1}^{n_{\max}} x_{ik}x_{jk}$. $Y_k \triangleq x_{ik}x_{jk} \in \{-1,1\}$ are identically distributed independent random variables for $i \neq j$. Let $E_2$ denote the event that $R_{i,j} \leq \frac{cd\log n}{1+\epsilon}$, $\forall j \neq i$, $\forall i$ which is irrelevant. Observe that for an irrelevant variable $i$, $\mathbb{E}[R(i,j)] = 0$ for any $j \neq i$. Thus, applying Lemma 4 with $a = \frac{n_{\max}}{(1+\epsilon)}$ for some constant $\epsilon > 0$ and $b = -1$ and $c = 1$, we have:

$$\Pr\left(E_2|E_1\right) = 1 - \Pr\left(\exists \text{ irrelevant variable } i, \ j \neq i : R(i,j) > a|E_1\right)$$

$$\geq 1 - n^2 \exp\left(-\frac{n_{\max}}{8(1+\epsilon)^2}\right) \ \text{(union bound)}$$

$$\geq 1 - O\left(\frac{1}{n}\right) \ \text{(for a large enough constant } c) \tag{7}$$

Therefore, when both $E_1$ and $E_2$ are true, then for all irrelevant variable $i$, $\mathcal{S}_i = i$. If $i$ is relevant, then $\mathcal{S}_i$ also contains variable(s) other than $i$. Now, for every $i$, $\mathcal{S}_i$ represents variables which participate in some hyperedge along with $i$ if $i$ is relevant, since for all such variable $j$, $R(i,j) = n_{max}$. Then, if $d$ is known, we take all possible $d$ subsets $M_{k_i}$ of $\mathcal{S}_i$ and correlate the corresponding parity function $M_{k_i}$ with the function values to find the coefficient. Since $m_1 = c2^k d\log n$ samples are available, error can be made less than $1/2^d$, and this gives an exact estimate with high probability when the result is rounded off to the nearest multiple of $1/2^d$. Let $E_3$ be the event that $\forall M_{k_i} \subset \mathcal{S}_i$, $\forall$ relevant $i$ : $|\sum_i f\left(\mathbf{x}_i\right)\chi_{M_{k_i}}\left(\mathbf{x}_i\right) - m_1\mathbb{E}[f\left(\mathbf{x}\right)\chi_{M_{k_i}}\left(\mathbf{x}\right)]| \leq m_1\frac{1}{2^d}$. Since, the function takes values between $0$ and $s$ (the number of hyperedges), taking $a = m_1\frac{1}{2^d}$, $b = -s$ and $c = s$, and applying Lemma 4, we have:

$$\Pr\left(E_3|E_1, E_2\right) \geq 1 - 2^k \exp\left(-\frac{m_1}{2^{2d+1}s^2}\right) \geq 1 - O\left(\frac{1}{n}\right).$$

Therefore, $\Pr\left(E_1 \bigcap E_2 \bigcap E_3\right) \geq 1 - O\left(\frac{1}{n}\right)$ concluding the proof of correctness for the algorithm.

There are at most $2^k$ parity functions to correlate. Thus the sample complexity of the algorithm is $m_1 = O(2^k d\log n + 2^{2d+1}s^2(\log n + k))$. The running time is $O(n^2 d\log n) + O(2^{2k} d\log n + 2^k 2^{2d+1}s^2(\log n + k))$. The first term in the running time is for forming the matrix $\mathbf{R}$. The second term in the running time is for correlation with $m_1$ samples for each of the $2^k$ parity functions.

**Remark:** Here, we have analyzed the algorithm in such a way that the first stage of forming $\mathbf{R}$ and thresholding using $n_{\max}$ only seems to tell us the relevant variables involved. In reality, running the algorithm yields $\mathbf{R}$ which after thresholding at $n_{\max}$ can identify distinct connected components and only the sub-structure of connected components has to be identified in the correlation step. Two variables are in the same connected component if they are in the same hyperedge. But our analysis is for the worst case when there is only one connected component. But it is possible to give a better bound in terms of the size of the largest component instead of $k$ (the total number of relevant variables). We do not pursue that in this proof.

## References

[1] S. Jukna, *Extremal Combinatorics*. Springer, 2011.