[Reviews · NeurIPS 2014]

Submitted by Assigned_Reviewer_7

The paper addresses the problem of learning sparse polynomials over the Boolean domain by introducing a computationally efficient algorithm. More specifically, assuming that a polynomial $f$, defined over the boolean domain ${ -1, 1}^n$, has at most $s$ nonzero coefficients, the paper proposes an algorithm to exactly reconstruct $f$ given random samples from the uniform distribution over the domain. The proposed algorithm, referred to as "LearnBool" has the low computational complexity of $O(n 2^s)$. The paper shows that LearnBool succeeds if $f$ satisfies a "unique sign pattern property", which states that each value of $f$ uniquely determines the sings of the parity functions. The paper shows that this property holds for the complement of the cut function on a hypergraph. Thus, the results in the paper can be used to learn the structure of the hypergraph from random cuts. The paper demonstrates the effectiveness of the proposed approach in terms of the computational time and performance on the Yahoo! messenger user communication pattern dataset.

The paper is well-organized, well-written and clear. The reviewer believes that the paper has sufficient novelty and makes an interesting contribution. In particular, by defining and exploiting the unique sign pattern property, the paper allows to reduce the list of potential nonzero coefficients of the polynomial from $O(2^n)$ to $O(2^s)$. As a result, this leads to significant reduction of the computational time from exponential in $n$ (as in the state of the art which are based on compressed sensing [12, 13]) to $O(n 2^s)$. The theoretical result guaranteeing that the the unique sign pattern property holds are quite interesting and as the paper shows in the particular case of hypergraph sketching this property always holds. The experimental results for hypergraph sketching on the Yahoo! messenger user communication pattern dataset are sufficient and convincing. In particular, in situations where $n$ is about a few hundreds and compressive sensing based methods cannot run due to high computational complexity, the algorithm runs in a matter of a few seconds (Table 1) and at the same time consistently achieves a low error rate (Figure 1).
Summary: The paper proposes a computationally efficient algorithm for the problem of learning sparse polynomials over the Boolean domain. The algorithm and the theory in the paper are quite interesting and experiments are convincing.

Submitted by Assigned_Reviewer_45

Summary.
This paper presents an efficient algorithm for learning sparse polynomial functions over boolean domain. The polynomial functions are represented by a linear combination of parity functions so that the learning is essentially finding all the non-zero coefficients. Given a set of observations, the sparse polynomial learning is formulated into a compressive sensing problem over a large scale linear system. Different from [12][13], this paper presents theoretical results on finding the active set (the indices of non-zero coefficients) efficiently by using the unique sign pattern property, and also an application to graph sketching in social network data.

Pros:
1. This paper presents novel theoretical results on finding the active set for sparse polynomial learning under reasonable conditions for both clean and noisy cases.
2. The empirical results in section 5 show that the proposed algorithm significantly improve the performance of graph sketching over compressive sensing algorithms.

Cons:
1. It will be great if the authors could provide a toy example to demonstrate the intuition behind the unique sign pattern property and the algorithm details.
2. In Algorithm 1, the sparsity parameter s seems critical. Would be possible to provide some heuristics to estimate it effectively?
3. Are there any situations that the unique sign pattern property does not hold? And how does the Algorithm 1 perform in these situations?
Summary: This paper presents a truly novel algorithm for efficient sparse polynomial learning with theoretical guarantee, which could have a major impact to many fields such as graphical models, sparse learning and social networks.

Submitted by Assigned_Reviewer_46

The paper describes an algorithm for learning sparse polynomials over the boolean domain. It differs from works like [12,13] in that even though it has the same sample-complexity, it has much better run-time. It uses standard l1 minimization techniques but is able to perform a "feature-selection" step to reduce the combinatorial number of subsets from 2^n to 2^s, where s is the sparsity of the polynomial.

Of all the 5 other paper I reviewed for NIPS, this was the best one. However, it was also the slowest for me to read and I am the least certain about it, mainly because it was the paper farthest from my own specialty. There were quite a few arguments that I couldn't check in detail and instead rely on trusting the authors.

The writing was clear, and the intuition was normally laid out well. I was at first disappointed that Lemma 2 and Theorem 4 were relegated to the appendix, but the appendix was short and I think this was OK.

Overall, assuming I didn't misinterpret anything, I'm enthusiastic about publishing this paper. There was more in this paper than in most of the other papers I reviewed combined. I am a bit confused why this paper is in NIPS and not STOC, though.

My minor criticisms are below:

- In the abstract, the description of the unique sign property was confusing, even after reading the definition in the paper. The definitions in the paper seemed good, but not the one in the abstract.

- I didn't notice the sample complexity mentioned in the first part of the paper. It is O(n), right? Please state this earlier, since it is of great interest (if it was 2^s, then a fast time complexity is sort of irrelevant)

- Theorem 2: parity functions are not yet defined, so this is not yet clear to a non-expert like myself. Similarly, section 2 was a good first-step, but I still needed to supplement this with some basic notes about Fourier expansion of boolean functions.

- Section 1.1. The difference in the results with [12,13] and [15] was made quite clear, but the difference in techniques was not. I am very curious how much of the technique was novel. Please state the key technical innovation (or was the main innovation just the discovery of the algorithm/idea, and then the analysis was standard?).

- Section 3 and later on, the notation for an ordered pair of samples and labels uses angular brackets, which are usually reserved for inner products. Why not just use parenthesis?

- Theorem 3 and following paragraph refer to feasible "solutions" of the problem, which is confusing since I interpret "solution" and "optimum" identically, and with this interpretation it makes no sense. I would say a "feasible point", not a "feasible solution".

- The technique reminds me of "Safe feature elimination" by El Ghaoui, who proves a similar reduction technique for l1 minimization (in the standard compressed sensing context), but of course quite different since his variables are not linked the same way yours are.

- The step of finding feasible solutions p in equation 4 in Algo 1 is not explained in the main text, so this is a bit of a mystery. It would be nice to give more intuition here. (Here, it is OK to call p a "solution" since you are just solving an equation, not solving an equation AND minimizing).

- Section 5 was either written by a different co-author or not proof-read to the same degree, because the writing quality deteriorates and there are many English mistakes. Bottom of page 5, "previously" is used twice in a sentence. There are many missing "the"'s in the sentences.

- After equation 6, what is "d"?

- Theorem 5 and this specialization is quite nice. However, it felt secondary so I did not check the corresponding section in the appendix (except to note that the proof of Thm 5 is hard to follow and condensed into 1 giant paragraph, very unlike the proof of Thm 4 and the lemma).

- 5.2, the dataset was described, but it was never said what exactly it is (for people not familiar with Yahoo! Messenger). Is this an instant messaging, or something else? It wasn't clear to me the connection between instant messaging and zip codes, and I think you could explain this in 1 or 2 sentences to clear it all up.

- In 5.2.1, you mention identifying the relevant variables correctly, but how do you know this? Was there a ground-truth solution? Did I miss something here?

- The references should capitalize "Fourier" in the titles of papers.
Summary: Strong paper with good new theory, but also an interesting application. Wasn't immediately clear if the application was more useful than existing approaches, but at least the approach seemed quite novel.
Author Feedback
Author rebuttal: We would like to thank the reviewers for their careful reviews and comments.

Reviewer_45:
1-We provide an example that would clarify the unique sign property (USP).
Example 1: Consider f1(x1, x2,..., xn) = x1-x1*x2+x3. For any input, the set of values [x1, x1*x2, x3] is a parity pattern. For example, if the input is [x1 = +1, x2 = -1, x3= +1], then the parity pattern is [+1, -1, +1]. For this input, the function has the value f [+1, -1, +1] = 3. Now the key is that 3 is the maximum value this function can obtain, and that this maximum is only obtained for the parity pattern [+1, -1, +1].

Example 2: A function that does not have USP: Consider f2(x1, x2,..., xn) = x1-x1*x2+x2. In this case, the maximum value of f2 is 1. Two inputs that produce this maximum are [x1 = +1, x2 = +1] and [x1 = +1, x2 = -1]. The parity pattern corresponding to these inputs are [+1, -1, +1] and [+1, +1, -1], respectively. Therefore there are multiple parity patterns that obtain the maximum function value, hence the USP does not hold.

How the unique sign property is used: To learn the function f1 that has USP, we use the fact that each parity of f1 attains the same value for all the inputs that yield f1 = 3. The algorithm has two main steps:

Conversion from Real to Boolean:
We convert -1 to 1 and 1 to 0 over the Boolean domain (GF(2)). Further, operation of multiplication is mapped to XORing (addition over GF(2)). When f1=3, x1*x2 = -1. This means that over GF(2), x1+x2 = 1. So every parity function gives a linear constraint over GF(2). All inputs (converted to GF(2)) that give f1 = 3, must satisfy this. The parity function x1*x2 can be represented as a vector of 1’s and 0s as p = [1 1 0 0 0 …] and [1 1 0 0 0 ...] [x1 x2 … xn]^T represents the value of the parity over the Boolean domain.

Solving Linear Equations over GF(2):
Since, all parities evaluate to 1 or -1 over all inputs giving the value 3, we form a matrix X that contains all such inputs. This has rank (at least) n-s over GF(2) since we know inputs satisfy at most one linear constraint imposed by each parity. We extract all possible parities from solving Xp=0 and Xp=1,using Gaussian elimination. Then we use only this set of candidate parities (at most 2^s) in the compressed sensing framework to solve the problem. The unique sign property coupled with domain conversion and solving linear equations reduces the complexity.

2-Our algorithm LearnBool does not require the knowledge of sparsity except when sampling. If sparsity is unknown, it is possible to choose a smaller sparsity and then run our algorithm to ‘test’ the output for consistency with the rest of the samples. If the sparsity level is incorrect this will fail with high probability. With this method we obtain a simple sparsity estimation procedure. It is possible that something smarter can be done but we do not investigate this in this work.

3-An example of such a function is in our reply to the first question. In such cases, our algorithm is not guaranteed to succeed.

Reviewer_46:
1-We will clarify the definition of the unique sign property in the abstract in the camera-ready version.
2-For the first part (Theorem 4), the sample complexity is O(n 2^s). The runtime complexity for learning is poly(n, 2^s). On the contrary, the previously known results (based on compressed sensing) had a running time O(2^n) but a better sample complexity O(ns^2). For any constant s, our sample complexity is linear in n. The second part gives a faster algorithm with better sample complexity for the special case of hypergraph sketching.
3-We agree that due to space limitations, we could not provide a more detailed background on the Fourier expansion of Boolean functions. We will expand the background section and also provide a reference for the non-expert readers in the camera-ready version.
4-The main innovation is the discovery of the unique sign property and how it leads to a significant reduction in the dimension of the candidate parity space. We refer the reviewer to our reply to question 1 of Reviewer_45.
5-Angular brackets is the standard notation for labeled input-output samples in the learning theory community. So we have adopted that to be consistent.
6-Feasible point is also a valid use as suggested by the reviewer. We would adopt this in our camera-ready version.
7-We thank the reviewer for pointing the connection out. We were not aware of the work of El Ghaoui. We will check if his technique can be useful in this context for our future work.
8-Solving Y_{max}p=0 and Y_{max}p=1 involves solving linear equations over GF(2) (Boolean Domain) using standard Gaussian Elimination (O(n^3) runtime). The matrix is rank deficient in this case. But the null space is of dimension at most s and therefore there are at most 2^s solutions to each of the equations which can be enumerated. We will make sure to mention this in the camera-ready version.
9-We are planning to go over Section 5 and improve the language and fix the errors you pointed out for the camera ready version.
10-d is the maximum size of a hyperedge in the hypergraph. It is defined in line 277.
11-We will definitely expand and clarify the proof of theorem 5 in the camera-ready version of the paper.
12-Yahoo messenger is an instant messaging application. The zip codes identify the geographical region that the messages are sent from (receiver zipcode also appears in the dataset). We will add 1 or 2 more sentences to clarify these points.
13-The relevant variables are the users that receive messages in a given time and space interval. Since we have the dataset and the selected zipcodes, we know which users receive the messages. The algorithm recovers the receivers correctly using the random c-cut function queries.
14-We thank the reviewer for pointing out the error. We will change “fourier” to “Fourier” in the references section.

Reviewer_7
We thank the reviewer for his kind comments and feedback.